# Improving Remote Sensing Image Super-Resolution Mapping Based on the Spatial Attraction Model by Utilizing the Pansharpening Technique

**Peng Wang [1,2,*], Gong Zhang [1], Siyuan Hao [3] and Liguo Wang [4]**

1. Key Laboratory of Radar Imaging and Microwave Photonics, Ministry of Education, Nanjing University of Aeronautics and Astronautics, Nanjing 210016, China; gzhang@nuaa.edu.cn
2. College of Computer Science and Technology, Nanjing University of Aeronautics and Astronautics, Nanjing 210016, China
3. College of Information and Control Engineering, Qingdao University of Technology, Qingdao 266520, China; lemonbananan@163.com
4. College of Information and Communication Engineering, Harbin Engineering University, Harbin 150001, China; wangliguo@hrbeu.edu.cn
* Correspondence: Pengwang-B614080003@hotmail.com; Tel.: +86-188-1621-0656

**Abstract:** The spatial distribution information of remote sensing images can be derived by the super-resolution mapping (SRM) technique. Super-resolution mapping, based on the spatial attraction model (SRMSAM), has been an important SRM method, due to its simplicity and explicit physical meanings. However, the resolution of the original remote sensing image is coarse, and the existing SRMSAM cannot take full advantage of the spatial–spectral information from the original image. To utilize more spatial–spectral information, improving remote sensing image super-resolution mapping based on the spatial attraction model by utilizing the pansharpening technique (SRMSAM-PAN) is proposed. In SRMSAM-PAN, a novel processing path, named the pansharpening path, is added to the existing SRMSAM. The original coarse remote sensing image is first fused with the high-resolution panchromatic image from the same area by the pansharpening technique in the novel pansharpening path, and the improved image is unmixed to obtain the novel fine-fraction images. The novel fine-fraction images from the pansharpening path and the existing fine-fraction images from the existing path are then integrated to produce finer-fraction images with more spatial–spectral information. Finally, the values predicted from the finer-fraction images are utilized to allocate class labels to all subpixels, to achieve the final mapping result. Experimental results show that the proposed SRMSAM-PAN can obtain a higher mapping accuracy than the existing SRMSAM methods.

**Keywords:** remote sensing image; super-resolution mapping; spatial attraction model; pansharpening technique

## 1. Introduction

Due to the variety of land-cover classes, and the limitations of sensors, there are many mixed pixels that exist widely in any original remote sensing image [1]. Although spectral unmixing [2] can handle mixed pixels by estimating the proportions of land-cover classes in mixed pixels, it cannot provide any spatial distribution information for remote sensing images. To solve this issue, Atkinson proposes a super-resolution mapping (SRM) technique, which is also named subpixel mapping [3,4]. SRM divides each mixed pixel into subpixels, and transforms the coarse-fraction images to a hard classification image with a higher spatial resolution [5].

In recent years, many studies on SRM have been rapid developed. The Hopfield neural network [6,7], back-propagation neural network [8,9], object spatial dependence [10,11], indicator cokriging (ICK) [12,13], point spread function [14,15], and some super-resolution methods [16–18] have been successfully utilized in SRM. The above methods belong to soft-then-hard super-resolution mapping (STHSRM) types. STHSRM contains two steps: (1) sub-pixel sharpening; and (2) class allocation [19]. When addressing a supervised classification problem, another type of algorithm, namely super-resolution then classification (STC) [20–22], can be utilized to obtain the spatial distribution of land-cover classes. The fine-resolution image is derived from the original coarse image by appropriate super-resolution reconstruction methods. An ideal result is then directly derived from the fine-resolution image by classification techniques. However, when there is no full supervision information in the classification process, STC is not always superior to STHSRM. So, STC is different from STHSRM. To optimize the mapping result, some artificial intelligence algorithms, such as particle swarm optimization [23,24], simulating annealing [25], and genetic algorithm [26], are utilized as the optimization model. In addition, various auxiliary information, such as sub-pixel-shifted images [27–29], light detection and ranging data [30], fused images [31], panchromatic images [32], and shape information [33] are used to improve the SRM performance.

Due to its simplicity, physical meanings, and no need for prior structure information, super-resolution mapping based on SRMSAM, belonging to the STHSRM type, has been widely applied. SRMSAM mostly differs in that the spatial attraction is computed, such as the subpixel/pixel spatial attraction model (SPSAM) [34], the subpixel/subpixel spatial attraction model (MSPSAM) [23], and the more effective hybrid spatial attraction model (HSAM) [35,36]. However, these SRMSAM methods are applied in the coarse-fraction images that are derived by unmixing the original coarse remote sensing image. Due to the coarse resolution of the original image, it is difficult for the coarse-fraction images to fully pick up the spatial–spectral information of the original image. To solve this issue, improving remote sensing image super-resolution mapping based on the spatial attraction model by utilizing the pansharpening technique (SRMSAM-PAN) is proposed. In SRMSAM-PAN, a novel processing path (pansharpening path) is added to the existing processing path. The pansharpening technique is utilized to fuse the original coarse remote sensing image with the high-resolution panchromatic image from the same area, to derive the improved image [37], and the novel fine-fraction images are obtained by unmixing the improved image. The two kinds of fine-fraction images from the pansharpening path and the existing processing path are then integrated to produce finer-fraction images with more spatial–spectral information. Finally, the values predicted from the finer-fraction images are used to allocate the class labels to each subpixel, to obtain the final mapping result. The experimental results show that the proposed SRMSAM-PAN produces a higher mapping accuracy than the state-of-the-art SRMSAM methods.

## 2. Theory of Spatial Correlation

The intention of SRM is to obtain the subpixel spatial distribution within mixed pixels, by maximizing their spatial correlation [3]. A simple example that explains the theory of the spatial correlation is shown in Figure 1. The original coarse remote sensing image contains two classes, representing Class A and Class B, respectively. The coarse-fraction image, which is shown in Figure 1a, has mixed pixels, and the proportion of Class A is marked on each mixed pixel. The zoom factor represents the zoom ratio between a mixed pixel and its subpixels. When the coarse-fraction image is upsampled with the zoom factor, each mixed pixel is segmented into 16 subpixels. A value of 0.25 means that the four subpixels are attributed to Class A in the central mixed pixel. Figure 1b,c describes two possible subpixel spatial distributions. Based on the theory of spatial correlation, the subpixel spatial correlation is greater in the same class. Therefore, Figure 1b is considered to be more optimal.

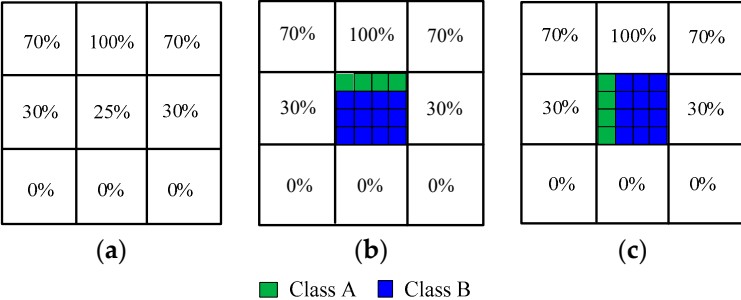

**Figure 1.** Example of spatial correlation. (**a**) Spectral unmixing result of Class A. (**b**) Probability of Distribution 1. (**c**) Probability of Distribution 2.

## 3. SRMSAM

The flowchart of the SRMSAM method is shown in Figure 2. Suppose that $S$ is the zoom factor, and that each mixed pixel is segmented into $S \times S$ subpixels. Firstly, $M$ coarse-fraction images $C_m$ ($m = 1, 2, \ldots, M$, $M$ is the number of land-cover class) are obtained by unmixing the original coarse remote sensing image. $C_m(P_N)$ is defined as the proportion value belonging to the $m$th class for the mixed pixel $P_N$ ($N = 1, 2, \ldots, K$), $K$ is the number of mixed pixels in the coarse-fraction image $C_m$). Secondly, the fine-fraction images $F_m$ are derived from the coarse-fraction images $C_m$ by the SRMSAM method. The fine-fraction images $F_m$ contain the predicted value $F_m(p_n)$, which is the predicted value of the $m$th class for subpixel $p_n(n = 1, 2, \ldots, KS^2)$, $KS^2$ is the number of subpixels in the mixed pixel $P_N$. The constraints should satisfy two conditions: (1) each subpixel is only assigned to a specific class, and (2) the number $L_m(P_N)$ of subpixels belonging to the $m$th class in the mixed pixel $P_N$ should meet Equation (1):

$$L_m(P_N) = Round(C_m(P_N) \times S^2) \tag{1}$$

where $Round(\bullet)$ is defined as a function that obtains the nearest integer to $C_m(P_N) \times S^2$.

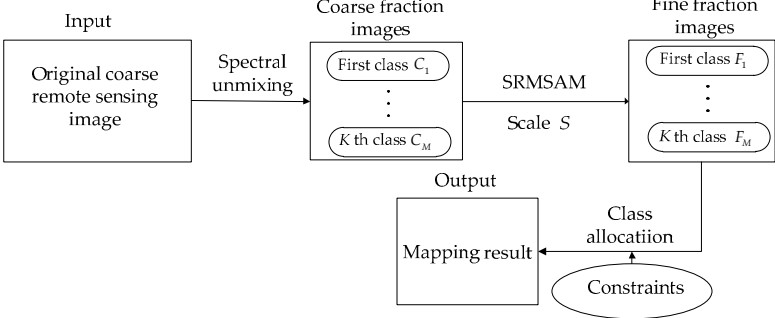

**Figure 2.** The flowchart of super-resolution mapping, based on the spatial attraction model (SRMSAM).

Finally, the predicted value $F_m(p_n)$ is used to allocate class labels into all subpixels by class allocation method.

In SRMSAM, the predicted value $F_m(p_n)$ can be computed by different spatial attraction models, such as the SPSAM model, MSPSAM model, and the more effective HSAM model. Next, we introduce the principles of the three models.

The SPSAM model considers the spatial correlation between the central subpixel $p_n$ and the neighboring pixel $P_J$ [34]. The predicted value $F_m^{\mathrm{spsam}}(p_n)$ of the SPSAM model can be written as:

$$F_m^{\mathrm{spsam}}(p_n) = \max \sum_{m=1}^{M} \sum_{n=1}^{KS^2} \sum_{J=1}^{8} o_{mn} \times w_n \times C_m(P_J) \tag{2}$$

$$o_{mn} = \begin{cases} 1, & \text{if subpixel } p_n \text{ belongs to class } m \\ 0, & \text{otherwise} \end{cases} \tag{3}$$

where $C_m(P_J)$ is the proportional value belonging to the $m$th class in the $J$th neighbor pixel $P_J$, and $J$ is the number of the neighbor pixels. In this paper, the number of the neighboring pixels is selected as eight [34]. $w_n$ is the weight for the dependence between the central subpixel $p_n$ and the neighboring pixel $P_J$:

$$w_n = \exp(-d(p_n, P_J)^2 / \varepsilon_1) \tag{4}$$

where $d(p_n, P_J)$ is defined as the Euclidean distance between the center subpixel $p_n$ and the coarse neighboring pixel $P_J$, which is shown in Figure 3a. $\varepsilon_1$ is the exponential model parameter.

In the MSPSAM model, the spatial correlation between the central subpixel and its neighboring subpixels is utilized to obtain the predicted value $F_m^{\text{mspsam}}(p_n)$ of the MSPSAM model [23]:

$$F_m^{\text{mspsam}}(p_n) = \max \sum_{n=1}^{KS^2} \sum_{j=1}^{8S^2} w_n \cdot x_{nj} \tag{5}$$

$$x_{nj} = \begin{cases} 1, & \begin{array}{l} \text{if subpixel } p_n \text{ and subpixel } p_j \text{ are} \\ \text{assigned to same land cover class} \end{array} \\ 0, & \text{otherwise} \end{cases} \tag{6}$$

where $w_j$ represents the weight of the spatial correlation between the central subpixel $p_n$ and the neighboring subpixel $p_j$, which is given as:

$$w_n = \exp(-d(p_n, p_j)^2 / \varepsilon_2) \tag{7}$$

As shown in Figure 3b, $d(p_n, p_j)$ stands for the Euclidean distance between the center subpixel $p_n$ to the neighboring subpixel $p_j$. $\varepsilon_2$ is the exponential model parameter.

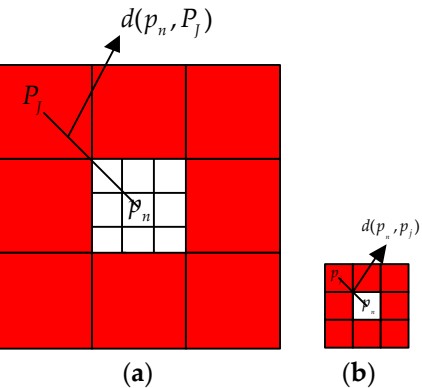

(a)      (b)

**Figure 3.** Euclidean distance. (**a**) Central subpixel $p_n$ and its eight neighboring pixels. (**b**) Central subpixel $p_n$ and its eight neighboring subpixels.

HSAM considers the spatial correlation of the above two models at the same time [35,36]. The predicted value $F_m^{\text{hsam}}(p_n)$ of the HSAM model can be derived by integrating the predicted value $F_m^{\text{spsam}}(p_n)$ of the SPSAM model, and the predicted value $F_m^{\text{mspsam}}(p_n)$ of the MSPSAM model by the appropriate parameter $\theta$:

$$F_m^{\text{hsam}}(p_n) = \theta F_m^{\text{mspsam}}(p_n) + (1-\theta) F_m^{\text{spsam}}(p_n) \tag{8}$$

Because the HSAM model inherits the advantages of the SPSAM model and the MSPSAM model, the final mapping result in the HSAM model outperforms the other two models.

## 4. Proposed Method

As shown in Figure 2, we can note that the existing SRMSAM methods are all applied in the coarse-fraction images derived from the original coarse remote sensing image. It is difficult for the coarse-fraction images to carry the full spatial–spectral information of the original image, due the coarse resolution in original image, the final mapping accuracy of SRMSAM will be affected. To supply more spatial–spectral information, and to improve the mapping accuracy, the SRMSAM-PAN model is proposed.

### 4.1. Pansharpening Path

In the SRMSAM-PAN model, a novel processing path (the pansharpening path) is added to the existing HSAM model. In the novel processing path, the resolution of the original coarse remote images is improved by fusing the higher spatial resolution panchromatic image from the same area by the pansharpening technique. The pansharpening technique can be considered as a particular data fusion problem, which aims at combining the spatial details from the panchromatic image and the spectral bands of the original remote sensing image. The improved image has a high spectral resolution of the original remote sensing image, and a high spatial resolution of the panchromatic image. Due to the effectively rendering of spatial details and its fast implementation, principal component analysis (PCA) [37] is selected as the pansharpening method in this paper.

Figure 4 gives the flowchart of the PCA pansharpening. Firstly, a set of scalar images called principal components is produced by a linear transformation of the original remote sensing image. The spatial information of the original image is collected in the first principal component, while the spectral information is concentrated in the other principal components. Subsequently, the spatial information from the high spatial resolution panchromatic image is utilized to replace the first principal component. To reduce spectral distortion in the PCA pansharpening processing, the histogram matching of the first principal component to the panchromatic image is performed before the replacement takes place. The histogram-matched panchromatic image shows the same mean and variance as the component to replace. Finally, the improved image is obtained by applying the inverse linear transformation. The mathematical model of the PCA pansharpening is given in Equation (9):

$$\hat{O}^b = \widetilde{O}^b + g_b(\text{PAN} - \text{I}) \tag{9}$$

where PAN is the panchromatic image $b$ ($b$ = 1, 2, ... , B), B is the number of spectral bands in the original image) is the $b$th spectral band, O is the original coarse remote sensing image; Ô is the improved image. $\hat{O}^b$ represents the $b$th spectral band of the improved image, $\widetilde{O}^b$ is the $b$th spectral band of the original image, which is interpolated at the scale of the panchromatic image, and $g_b = [g_1, g_2, ... , g_B]$ is the vector of the injection gains., while I is given as:

$$\text{I} = \sum_{b=1}^{B} y_b \widetilde{O}^b \tag{10}$$

where the weight vector $y_b = [y_1, y_2, ... , y_B]^{\text{T}}$ measures the spectral overlap between the panchromatic image and the spectral bands.

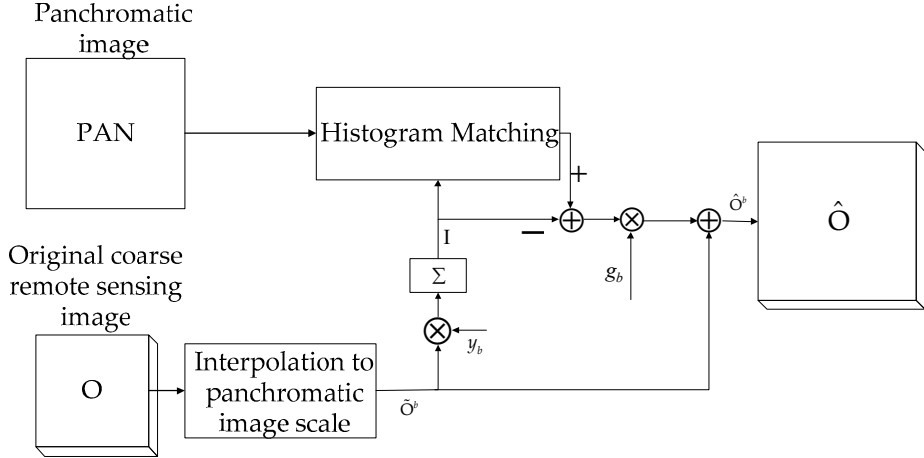

**Figure 4.** The flowchart of the principal component analysis (PCA) pansharpening.

### 4.2. Implementation of SRMSAM-PAN

In the pansharpening path, the improved image $\hat{O}$ is unmixed to produce the novel fine-fraction images $F_m^{pan}$, which contain the predicted value $F_m^{pan}(p_n)$. As shown in the flowchart of the pansharpening path in Figure 5, the novel fine-fraction images $F_m^{pan}$ with the predicted value $F_m^{pan}(p_n)$ are derived in two steps. In the first step, the original coarse remote sensing image O is improved to obtain an improved image $\hat{O}$, by the pansharpening technique. The second step is that the fine-fraction images $F_m^{pan}$ are obtained by directly unmixing the improved image $\hat{O}$. The predicted value $F_m^{pan}(p_n)$ of the land-cover classes in the fraction images is calculated by Equation (11).

$$V_b^{\hat{O}} = E \cdot F_m^{pan} + n \tag{11}$$

where $V_b^{\hat{O}} = [V_1^{\hat{O}}, V_2^{\hat{O}}, \ldots, V_B^{\hat{O}}]^T$ is the vector of the spectral value of the improved image $\hat{O}$, B are the number of spectral bands, $F_m^{pan} = [F_m^{pan}(p_1), F_m^{pan}(p_2), \ldots, F_m^{pan}(p_{KS^2})]^T$ is the vector of the predicted value $F_m^{pan}(p_n)$ of land-cover classes, $KS^2$ is the number of subpixels, E is the matrix for spectral endmembers, and n is the random noise. In this paper, the least squares support vector machine model (LSVM) [38] is used to seek the optimal estimation under the condition of the minimum random noise. Since the resolution of the original coarse remote sensing image is improved, the novel fine-fraction images can contain the more spatial–spectral information from the original image.

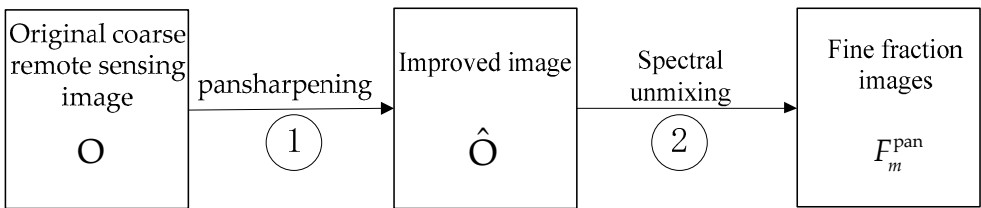

**Figure 5.** The flowchart of the pansharpening path.

Next the novel fine-fraction images with the predicted value $F_m^{pan}(p_n)$ from the pansharpening path, and the existing fine-fraction images with the predicted value $F_m^{hsam}(p_n)$ from the existing HSAM model are integrated to produce the finer-fraction images with a more accurately predicted value $F_m(p_n)$. The integrating process of the equation is given as:

$$F_m(p_n) = \alpha F_m^{pan}(p_n) + (1 - \alpha) F_m^{hsam}(p_n) \tag{12}$$

where $\alpha$ ($0 \leq \alpha < 1$) is the weight parameter to balance the influence of the predicted values, $F_m^{pan}(p_n)$ and $F_m^{hsam}(p_n)$. The class allocation method utilizes the more accurately predicted value $F_m(p_n)$

to allocate the class labels to each subpixel, to derive the final mapping result. The flowchart of SRMSAM-PAN is shown in Figure 6. The implementation of SRMSAM-PAN can be summarized in the following steps.

Step 1. In the existing path, the coarse-fraction images are derived from the original coarse remote sensing image by spectral unmixing. At the same time, the resolution of the original image is improved by the pansharpening technique in the pansharpening path.

Step 2. The fine-fraction images with the predicted value $F_m^{\text{hsam}}(p_n)$ are produced by the HSAM model. Also, the novel fine-fraction images with the predicted value $F_m^{\text{pan}}(p_n)$ are derived by unmixing the improved image.

Step 3. The fine-fraction images from the existing path, and the fine-fraction images from the novel pansharpening path are integrated to produce finer-fraction images with a more accurate predicted value $F_m(p_n)$ (see Equation (12)).

Step 4. According to the constraints in Equation (1), the more accurate predicted value $F_m(p_n)$ is used to allocate class labels to each sub-pixel for obtaining the final mapping result.

Comparing Figure 2 with Figure 6, the two kinds of fine-fraction images from the two paths are related to the different predicted values. The more spatial–spectral information that is supplied by the pansharpening technique, the higher the mapping accuracy that is generated by the proposed SRMSAM-PAN.

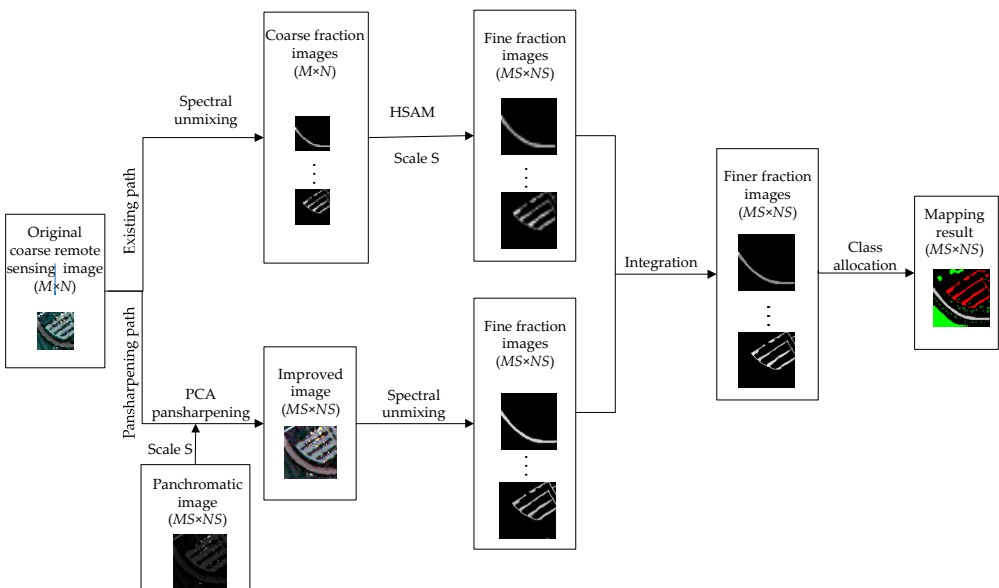

**Figure 6.** The flowchart of improving remote sensing image super-resolution mapping based on the spatial attraction model by utilizing the pansharpening technique (SRMSAM-PAN).

## 5. Experiments and Analysis

Three real hyperspectral images were implemented to test the performance of the proposed SRMSAM-PAN. To assess the effect of image registration error on the SRMSAM methods, a simulated coarse remote sensing image was produced by downsampling the original fine hyperspectral image [39]. The spectral response function of the IKONOS satellite was used to produce a suitable panchromatic image, in order to only consider the influence of the pansharpening technology on the mapping result, and to avoid the impact of errors caused by the acquisition of the panchromatic image [40]. The highest soft-attribute values assigned first (HAVF) [41] were considered as the class allocation method. All experiments were tested by the MATLAB 2018a software package (https://www.mathworks.com/).

Four SRMSAM methods were tested and compared: SPSAM [34], MSPSAM [23], HSAM [36], and the proposed SRMSAM-PAN. The mapping accuracy was evaluated quantitatively by the overall accuracy (OA) and the Kappa coefficient (Kappa).

*5.1. Experiment 1*

The first hyperspectral image was acquired over the Engineering School at the University of Pavia [39]. As shown in Figure 7a, the tested region had $100 \times 100$ pixels, 103 spectral bands, and 1.3 m spatial resolution. Figure 7a is degraded, with $S = 4$, to simulate the coarse image shown in Figure 7b. The panchromatic image shown in Figure 7c is produced by the spectral response of the IKONOS satellite. As shown in Figure 7d, the pansharpening technique was utilized to fuse the coarse remote sensing image and the panchromatic image, to produce the improved image. The improved image had the spectral resolution of the former, and the spatial resolution of the latter. As a visual observation, the improved image was similar to the original image. SRMSAM-PAN can use the improved image to obtain a higher mapping accuracy. The weight parameter $\alpha$ was set to 0.6.

As shown in Figure 8a, the reference image contained asphalt, meadows, trees, and bricks. The SRMSAM results of the four methods were given in Figure 8b–e. SRMSAM-PAN obtained a better mapping result than SPSAM, MSPSAM, and HSAM, by visual comparison. For example, there were many disconnected patches and obvious burrs in the bricks. This phenomenon was alleviated with the aid of the pansharpening technique. Due to supplying more spatial–spectral information, SRMSAM-PAN was closer to the reference image than the other three SRMSAM methods.

For quantitative evaluation, we utilized the mapping accuracy (%) of each class and OA (%), to evaluate the performance of the four methods. Checking the evaluation results shown in Table 1, the mapping accuracy of SRMSAM-PAN was higher than the other three methods. For example, the mapping accuracy (%) of trees in the SRMSAM-PAN increased from 56.32% to 72.31%, when compared with the HSAM. Since more spatial–spectral information was supplied by the pansharpening technique, SRMSAM-PAN could achieve the highest OA of 93.87%.

**Table 1.** Mapping accuracy (%) of the four methods ($S = 4$).

|         | SPSAM | MSPSAM | HSAM  | SRMSAM-PAN |
|---------|-------|--------|-------|------------|
| Meadows | 96.37 | 97.10  | 97.73 | 99.13      |
| Asphalt | 95.48 | 97.29  | 97.47 | 99.82      |
| Tress   | 45.13 | 55.23  | 56.32 | 72.31      |
| Bricks  | 77.18 | 83.37  | 83.60 | 90.30      |
| OA      | 85.17 | 88.73  | 89.20 | 93.87      |

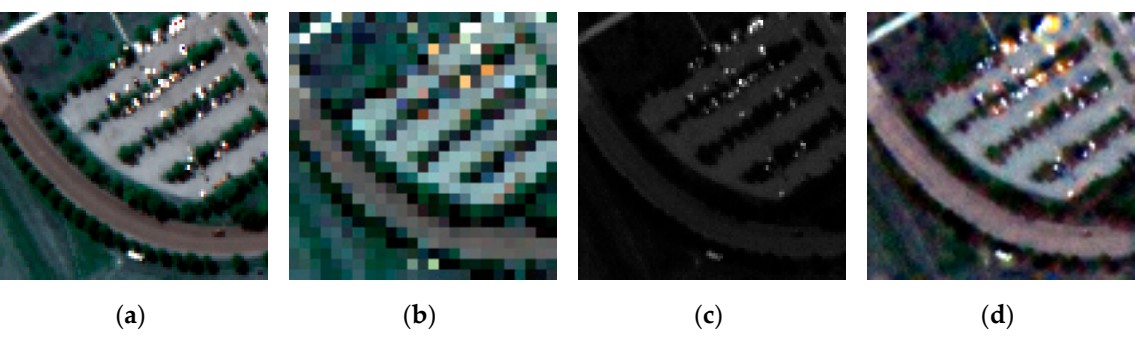

| (a) | (b) | (c) | (d) |

**Figure 7.** (**a**) RGB composites of images (bands 19, 30, and 44 for red, green, and blue, respectively). (**b**) Coarse image ($S = 4$). (**c**) Panchromatic image. (**d**) Pansharpening result.

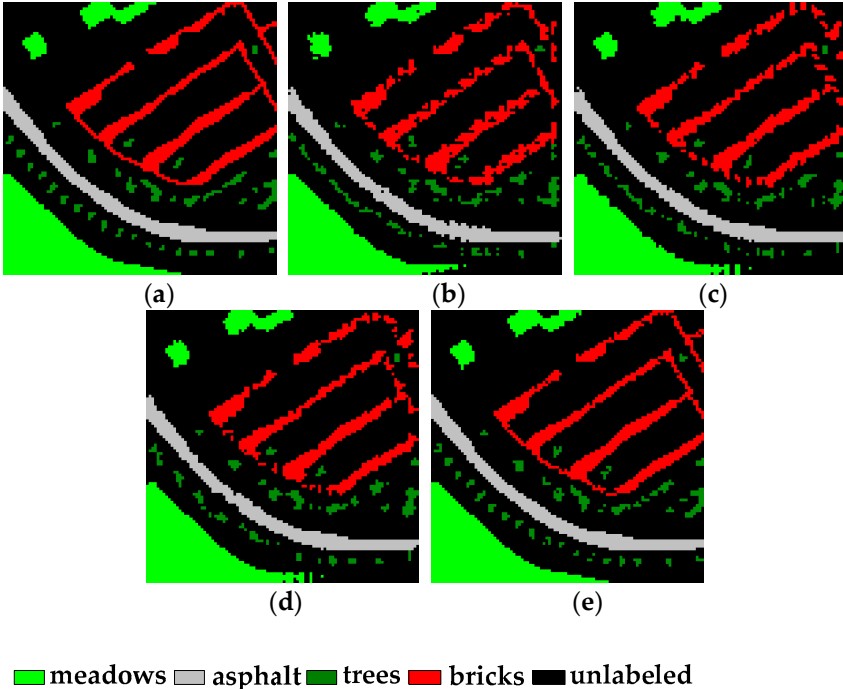

meadows ▭ asphalt ▬ trees ▬ bricks ▬ unlabeled

**Figure 8.** SRMSAM results in Experiment 1 ($S = 4$). (**a**) Reference image. (**b**) Subpixel/pixel spatial attraction model (SPSAM). (**c**) Subpixel/subpixel spatial attraction model (MSPSAM). (**d**) Hybrid spatial attraction model (HSAM). (**e**) SRMSAM-PAN.

*5.2. Experiment 2*

In Experiment 2, a hyperspectral image with 102 spectral bands and 1.3 m spatial resolution, which was larger and contained more classes, was used [42]. As shown in Figure 8a, the tested region, which covers the two residential areas on both sides of the Ticino river in Pavia city, had $400 \times 400$ pixels. Figure 9a is degraded, with $S = 4$, to obtain the simulated coarse image shown in Figure 9b. Figure 9c is the panchromatic image by the method described in Experiment 1. The pansharpening result is shown in Figure 9d. Better SRMSAM results were obtained by using more spatial–spectral information from the pansharpening result. The weight parameter $\alpha$ was set to 0.5.

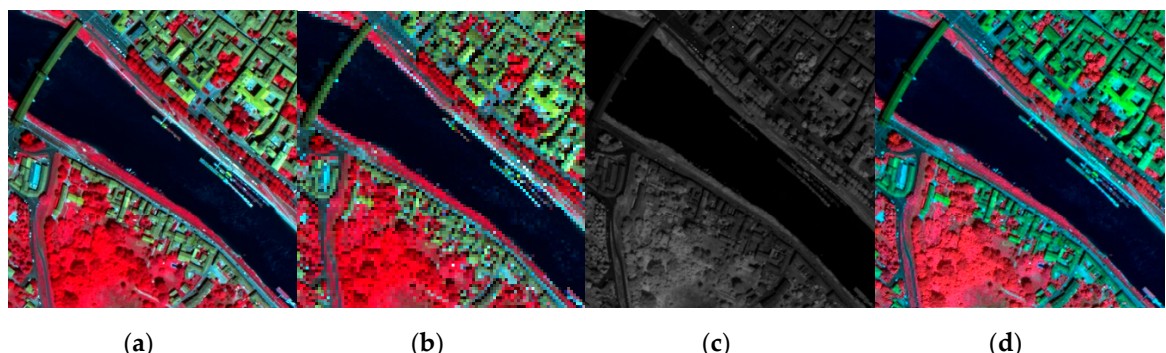

**Figure 9.** (**a**) RGB composites of image (bands 102, 56, and 31 for red, green, and blue, respectively). (**b**) Coarse image ($S = 4$). (**c**) Panchromatic image. (**d**) Pansharpening result.

The reference image in Figure 10a showed six classes, containing shadow, water, road, tree, grass, and roof. Figure 10b–e gives the mapping results of four methods. With the help of the pansharpening technique, the mapping result was more continuous, and the boundaries were smoother in Figure 10e. The SRMSAM-PAN was visually closer to the reference image, compared to the other three SRMSAM methods.

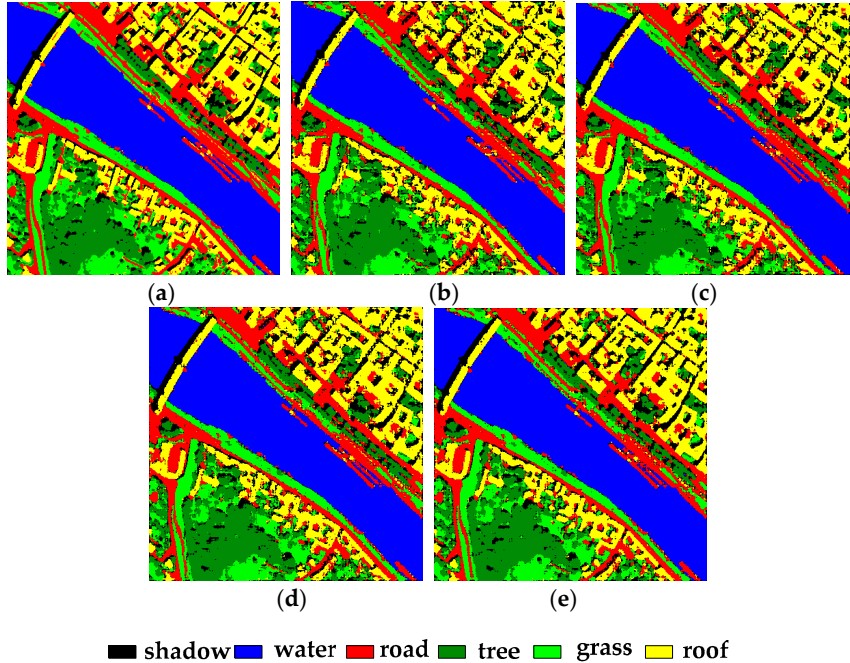

(a)  (b)  (c)

(d)  (e)

■ shadow ■ water ■ road ■ tree ■ grass ☐ roof

**Figure 10.** SRMSAM results in Experiment 2 ($S = 4$). (**a**) Reference image. (**b**) SPSAM. (**c**) MSPSAM. (**d**) HSAM. (**e**) SRMSAM-PAN.

The mapping accuracies (%) of each class and of the OA (%) of the four methods were measured in Table 2 Similar to the result in Experiment 1, both the mapping accuracies of each class and the OA (%) in the SRMSAM-PAN were higher than in the other three methods. In addition, to test the influence of the zoom factor $S$ on the final mapping results, the four methods were experimented for the two other zoom factors of 2 and 8. Figure 11a,b show the OA (%) and Kappa of the four methods for the three zoom factors. We can note that no matter how the zoom factor $S$ changed, the OA (%) and Kappa of the SRMSAM-PAN result were higher than SPSAM, MSPSAM, and HSAM.

**Table 2.** Mapping accuracy (%) of the four methods ($S = 4$).

|  | SPSAM | MSPSAM | HSAM | SRMSAM-PAN |
|---|---|---|---|---|
| Shadow | 52.46 | 62.80 | 65.98 | 74.57 |
| Water | 98.04 | 98.33 | 98.35 | 98.76 |
| Road | 79.38 | 82.97 | 84.03 | 89.74 |
| Tree | 80.95 | 83.47 | 84.52 | 89.00 |
| Grass | 80.51 | 83.94 | 85.66 | 89.41 |
| Roof | 85.89 | 88.63 | 89.87 | 92.49 |
| OA | 88.52 | 90.86 | 92.20 | 95.11 |

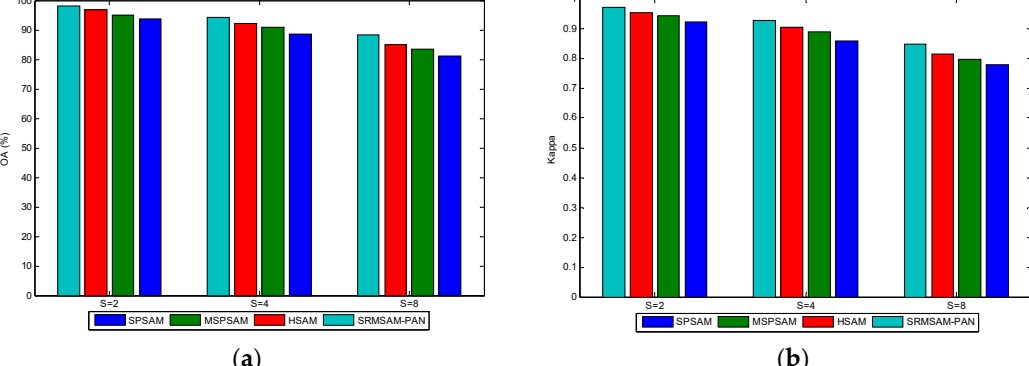

(a)  (b)

**Figure 11.** (**a**) Overall accuracy (OA (%) of the four methods in relation to the zoom factor $S$. (**b**) Kappa coefficient (Kappa) of the four methods in relation to the zoom factor $S$.

## 5.3. Experiment 3

The third dataset, including 191 bands and 3 m spatial resolution, was collected from a mall in Washington, DC [42]. As shown in Figure 12a, there were 240 × 280 pixels in the region tested. The coarse image shown in Figure 12b is obtained by downsampling Figure 11a, with $S = 4$. Figure 12c,d were the panchromatic image and the pansharpening results, respectively. The weight parameter $\alpha$ was selected as 0.5.

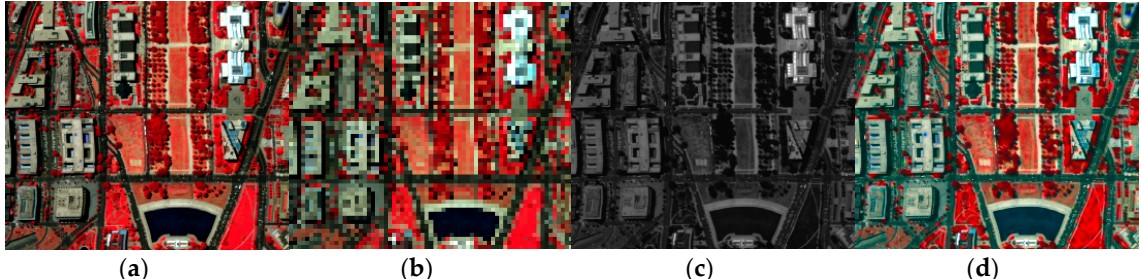

| (**a**) | (**b**) | (**c**) | (**d**) |

**Figure 12.** (**a**) RGB composites of the image (bands 102, 56, and 31 for red, green, and blue, respectively). (**b**) Coarse image ($S = 4$) (**c**) Panchromatic image. (**d**) Pansharpening result.

There were seven classes, including shadow, water, road, tree, grass, roof, and trail, in the reference image shown in Figure 13a. The SRMSAM results of the four methods are listed in Figure 13b–e. We can note there were many speckle artifacts existing in Figure 13b,c. Due to the pansharpening technique supplying more spatial–spectral information, this phenomenon was improved, and the SRMSAM-PAN result was more similar to the reference image.

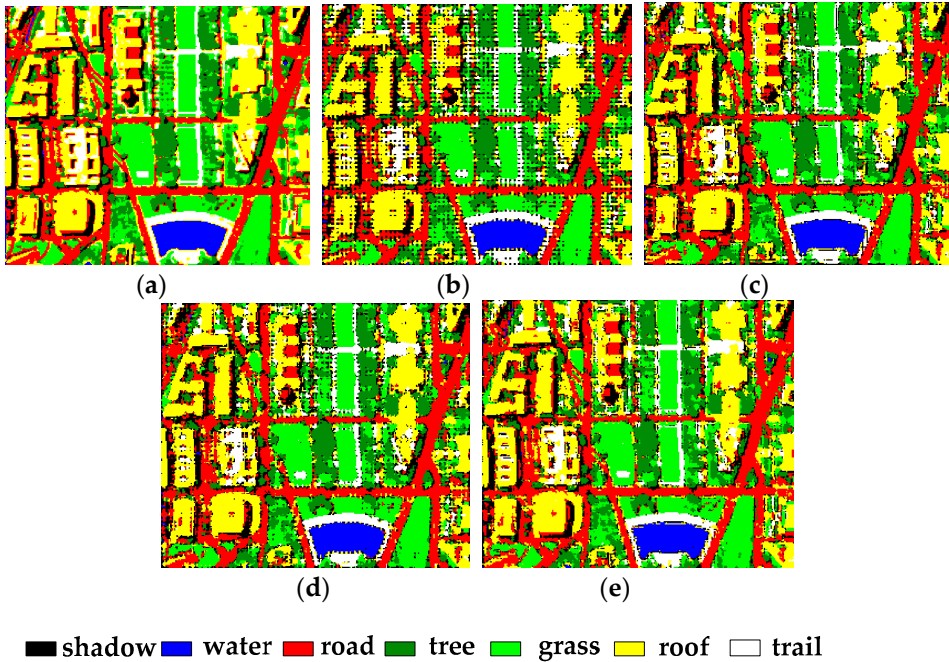

■ shadow ■ water ■ road ■ tree ■ grass ☐ roof ☐ trail

**Figure 13.** SRMSAM results in Experiment 3 ($S = 4$). (**a**) Reference image. (**b**) SPSAM. (**c**) MSPSAM. (**d**) HSAM. (**e**) SRMSAM-PAN.

Consistent with the results in Experiment 1 and Experiment 2, we also evaluated the four SRMSAM methods for three zoom factors, i.e., 2, 4, and 8. Checking the OA (%) and Kappa in Figure 14a,b, the quantitative evaluation from SRMSAM-PAN was higher than the other three SRMSAM methods.

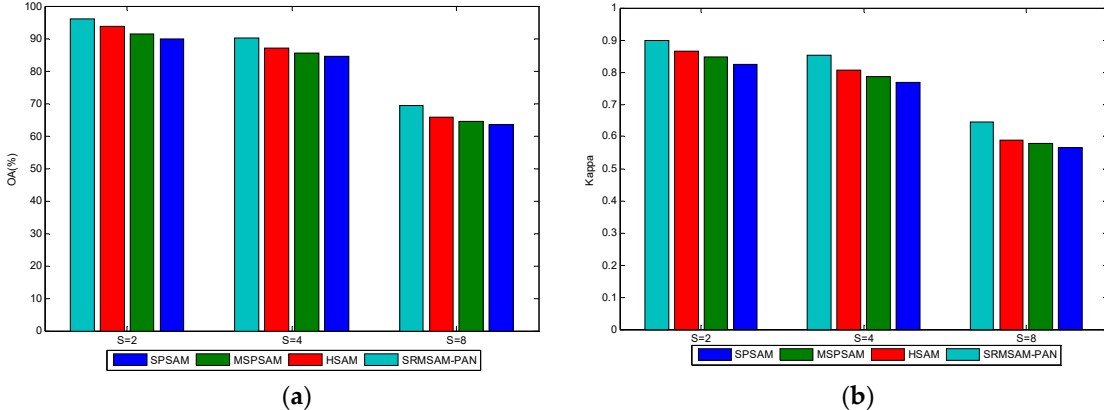

**Figure 14.** (**a**) OA (%) of the four methods in relation to the zoom factor *S*. (**b**) Kappa of the four methods in relation to the zoom factor *S*.

## 5.4. Discussion

The weight parameter $\alpha$ ($0 \leq \alpha < 1$) was utilized to balance the influence of the predicted values, $F_m^{\text{pan}}(p_n)$ and $F_m^{\text{hsam}}(p_n)$, on the proposed SRMSAM-PAN. To find the appropriate weight parameter $\alpha$, the experimental data, such as Experiment 2 ($S = 4$) and Experiment 3 ($S = 4$) calculated the adjusted OA (%) values for 10 combinations of $\alpha$ in the range of [0, 0.9] at intervals of 0.1. The method of selecting the weight parameter in Experiment 1 was the same as that in the Experiment 2 and Experiment 3. As shown the test result in Figure 15a,b, when $\alpha = 0$, only the HSAM model worked, and the predicted value $F_m^{\text{pan}}(p_n)$ from the pansharpening path did not play any role at this time. As $\alpha$ continued to increase, the OA (%) value was obviously improved. This was because of the greater spatial–spectral information that was supplied by the pansharpening path. When the appropriate $\alpha$ was 0.5 in Experiment 2 and Experiment 3, the pansharpening path produced the most spatial–spectral information, resulting in the highest OA (%). However, when $\alpha$ continued to increase, the predicted value $F_m^{\text{hsam}}(p_n)$ from the existing path contributed less to the overall solution (Equation (12)). This loss of information from the existing path degraded the value of the OA (%).

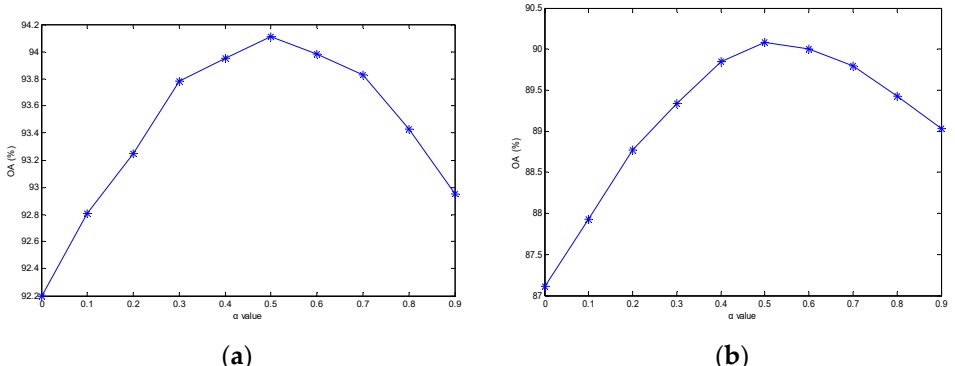

**Figure 15.** (**a**) OA (%) of SRMSAM-PAN in relation to the weight parameter $\alpha$ in Experiment 2 ($S = 4$). (**b**) OA (%) of SRMSAM-PAN in relation to the weight parameter $\alpha$ in Experiment 3 ($S = 4$).

In addition, the final mapping result also could be obtained by pansharpening technique then classification (PTC) which belongs STC type. Although the SRMSAM-PAN and PTC belong to different types, a comparison between STHSRM-PAN and PTC is worth studying. To get a fair comparison, the classification method based on SVM was selected in PTC. The number of the training samples was respectively selected as 30%, 20% and 10% per class, the remaining numbers per class were test samples in SVM. We respectively named the PTC with 30% training samples as PTC1, the PTC with 20% training samples as PTC2 and the PTC with 10% training samples as PTC3. The proposed STHSRM-PAN and

the three kinds of PTC methods were compared in Experiment 2 and Experiment 3. As shown the test result in Figure 16a,b, it is note that when there was abundant supervisory information (i.e., training samples), PTC is superior to the proposed STHSRM-PAN. Instead, STHSRM-PAN can obtain the higher OA (%) than PTC in the absence of adequate supervisory information. However, supervisory information is usually acquired by human markers, a large amount of supervisory information is often difficult to obtain. Therefore, STHSRM-PAN is more widely used than PTC for coarse remote sensing image.

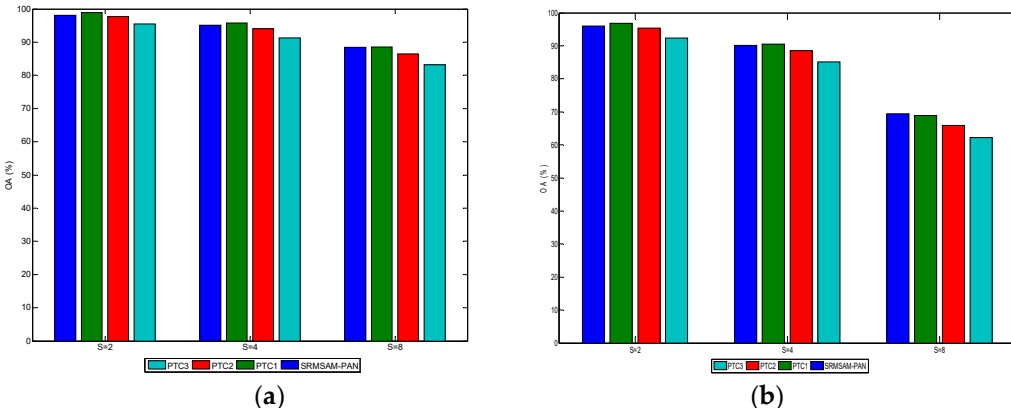

**Figure 16.** (**a**) OA (%) of SRMSAM-PAN and pansharpening technique then classification (PTC) in experiment 2 (**b**) OA (%) of SRMSAM-PAN and PTC in experiment 3.

Finally, the performance of SRMSAM-PAN depended on the pansharpening technique. Therefore, it was necessary to test the effects of different pansharpening methods on the performance of the proposed method. The band-dependent spatial detail (BDSD) [43] was selected as another pansharpening method to compare the previous PCA in Experiment 2 and Experiment 3 for three zoom factors, that is, 2, 4 and 8. Figure 17a,b show the OA (%) of the SRMSAM-PAN result in relation to the two pansharpening methods. As shown in Figure 17a,b, since BDSD is more effective than PCA [37], the OA (%) in the BDSD-based SRMSAM-PAN is higher than that in PCA-based SRMSAM-PAN. Hence the more effective pansharpening method can obtain a better mapping result.

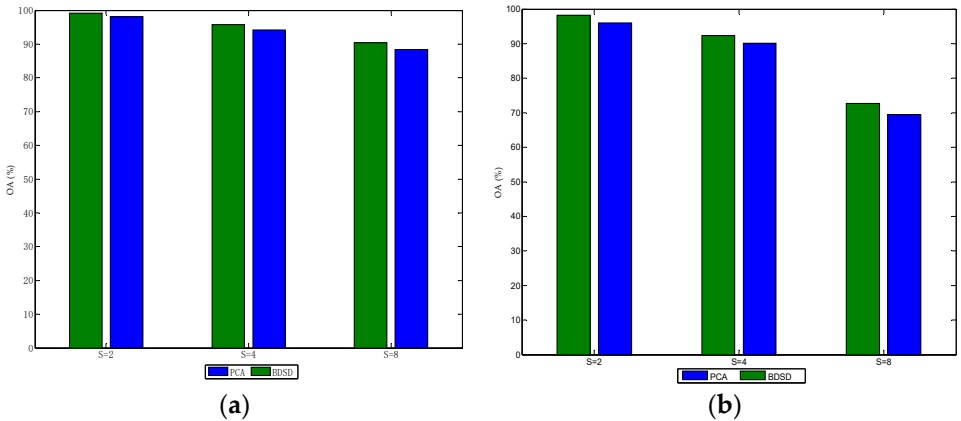

**Figure 17.** (**a**) OA (%) of the SRMSAM-PAN results in relation to the band-dependent spatial detail (BDSD) and PCA in Experiment 2. (**b**) OA (%) of the SRMSAM-PAN result in relation to BDSD and PCA in Experiment 3.

## 6. Conclusions

The contribution of this research is to improve the existing super-resolution mapping based on spatial attraction by pansharpening technique, resulting in obtaining a more accurate super-resolution

mapping result. In the proposed SRMSAM-PAN, first the pansharpening technique was utilized, to improve the resolution of the original image in the novel pansharpening path, and the novel fine-fraction images were obtained by unmixing the improved image. The finer-fraction images with more spatial–spectral information are derived by integrating the novel fine–fraction images and the existing fine-fraction images. Finally, the final mapping result was produced by the class allocation method, according to the values predicted from the finer-fraction images. The experimental results show that the proposed SRMSAM-PAN with the appropriate parameter obtained a better mapping result, compared with the three SRMSAM methods: SPSAM, MSPSAM, and HSAM.

Since the performance of the proposed SRMSAM-PAN is related to the pansharpening technique, a better mapping result can be obtained by the more effective pansharpening method. It is worth in developing a more effective pansharpening method in the future. Moreover, the appropriate parameter $\alpha$ is selected by multiple tests in this paper. To improve the final mapping result, an adaptive method for selecting the most appropriate weight parameter $\alpha$, is worth studying in future work. Finally, we simulate the coarse remote sensing image by downsampling the original fine image. Hence, the performance of the proposed method in the real coarse remote sensing image will be further studied.

**Author Contributions:** Conceptualization, P.W.; Methodology, P.W.; Software, G.Z.; Validation, G.Z., S.H. and L.W.; Formal analysis, S.H.; Investigation, G.Z.; Resources, L.W.; Data curation, P.W.; Writing—original draft preparation, P.W.; Writing—review and editing, G.Z.; Visualization, L.W.; Supervision, G.Z.; Project administration, P.W.; Funding acquisition, P.W.

**Funding:** The work was supported by the National Natural Science Foundation of China (grant no. 61801211, 61871218, 61701272, 61675051), Fundamental Research Funds for the Central University (grant no. 3082017NP2017421), National Aerospace Science Foundation of China (grant no. 20185152).

**Acknowledgments:** The authors would like to thank Dr. Qunming Wang of Tongji University for providing the relevant data set. The authors would like to thank the handling editors and the reviewers for providing valuable comments.

**Conflicts of Interest:** The authors declare no conflict of interest.

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
