# Peer review of "Improving Remote Sensing Image Super-Resolution Mapping Based on the Spatial Attraction Model by Utilizing the Pansharpening Technique"

_remotesensing, doi:10.3390/rs11030247_

Round 1
Reviewer 1 Report
The paper is fine but some significant English proofreading is required.
The authors also need to:
explain the software used for the analysis
improve the discussion on the research gaps.
add more recent references.
Author Response
Dear reviewer
We greatly appreciate your time on our paper and clear instructions for revision.
We have carefully considered your comments and suggestion. We provided item-by-item response. The whole paper has been revised accordingly and the revised parts are marked in green. We hope the new manuscript will meet your standard. The attachment is response. Moreover, this manuscript has been carefully edited by MDPI English Editing Service, and the grammar, spelling, and vocabulary have been improved where needed.
In the end, we want to convey our earnest thanks to you. Your valuable comments have greatly improved this manuscript.
Best wishes
The authors

Reviewer 2 Report
This article present capabilities to improve the remote sensing image analysis and focus on mixed pixels. The goal is to enhance super-resolution mapping thank to pansharpening technique.
This article starts to give a simple example to subpixel image by spacial correlation. Then it gives the flowchart of SRMSAM and the three ways to compute its values. It explains the present method don't take in account the higher resolution carried by panchromatic image. The proposition is to produced prediction only with pansharpening technique, then combined the result with HSAM. This article presents some results. The experiments take place at Pavia Center, Pavia River and Washington, with significant improvement each time. The article finally draws perspectives.
The document structure is normally balanced : introduction 35 lines, correlation 24 lines, srmsam 43 lines, method 67 lines, results 122 lines, future works 12 lines.
34 references from 1997 to 2019 (20% before 2010, 30% after 2015)
Remarks
The explanations are clear. The document is easy to read. Readers can be convinced of thecapabilities to reproduce the experiments.
L144, figure 4: an arrow is missing from block “O” to block “Interpolation to panchromatic image scale”.
L310: Why choosing alpha = 0.6 in experiment 1 (L216) and showing here the best value whith 0.5?
L326: In my opinion, the article doesn't give any clue to understand why OA increase when alpha goes to 0.5 and why it decreased after. Furthermore, there is no suggestion about merging pansharping and unmixing processing.
L376, L384: years are not in bold
Author Response
Dear reviewer
We greatly appreciate your time on our paper and clear instructions for revision.
We have carefully considered your comments and suggestion. We provided item-by-item response. The whole paper has been revised accordingly and the revised parts are marked in green. We hope the new manuscript will meet your standard. The attachment is response.
In the end, we want to convey our earnest thanks to you. Your valuable comments have greatly improved this manuscript.
Best wishes
The authors

Reviewer 3 Report
Dear Authors: From my point of view, the quality and novelty of the paper is adequate, however requieres to improves some topics.
For me is mandatory to include as a results a classification of fused image, in order to evluate the impact of the SRM compared with a "simple" pansharpened.
Considering that the greatest contribution of work this is to include the pansharpning path in the SRM, it is necessary to include more than one pansharpening method in order to evaluate the impact of this issue in the final results
Must be included a recomendation or methodology to determine a range (at least) of alpha value
In equation 8 change H to F
Author Response

(The authors gave the same response as above.)

Round 2
Reviewer 3 Report
The manuscript has been improved in a good way. However, according to the aim of the work, the conclusion are weak. I recommend to the authors improves this section to allows readers a clear idea of this research and its scopes.
For example, the sentence written on page 13, Line 371 "Finally, the performance of SRMSAM-PAN depended on the pansharpening technique." could be developed as an important conclusion
Page 8 Line 233-234: It might be important for future readers to have a WEB address from which to download the software package for testing purposes. This significantly increases the impact of your work, and makes it more visible to the scientific community.
Author Response
Dear Reviewer,
Thank you for clear instructions of revision. We have carefully considered your comments and suggestion. We have carefully provided item-by-item response to your comments. The revised parts are marked in green. The attachment is the response. Thanks a lot.
Best wishes
The authors
